# STABILIZING KNOWLEDGE, PROMOTING REASONING: DUAL-TOKEN CONSTRAINTS FOR RLVR

## ABSTRACT

Reinforcement Learning with Verifiable Rewards (RLVR) has become an effective post-training method for improving the reasoning abilities of Large Language Models (LLMs), mainly by shaping higher-order behaviors such as reflection and planning. However, previous RLVR algorithms often apply uniform training signals to all tokens, without considering the different roles of low-entropy **knowledge-related tokens** and high-entropy **reasoning-related tokens**. Some recent methods try to separate these token types by gradient masking or asynchronous updates, but these approaches may break semantic dependencies in the model output and hinder effective learning. In this work, we propose **Archer**, an entropy-aware RLVR approach with **dual-token constraints** and synchronous updates. Specifically, our method applies weaker KL regularization and higher clipping thresholds to reasoning tokens to encourage exploration, while using stronger constraints on knowledge tokens to maintain factual knowledge. Experimental results on several mathematical reasoning and code generation benchmarks show that our approach significantly outperforms previous RLVR methods, reaching or exceeding state-of-the-art performance among models of comparable size.

## 1 INTRODUCTION

Large Language Models (LLMs) have shown strong capabilities across various domains, as demonstrated by models like OpenAI's "o" series (OpenAI, 2024a;b) and DeepSeek-R1 (DeepSeek-AI et al., 2025). While supervised pre-training enables LLMs to acquire vast amounts of world knowledge, post-training techniques such as Reinforcement Learning (RL) (DeepSeek-AI et al., 2025; Kimi Team et al., 2025; Yang et al., 2025a) and test-time scaling (Snell et al., 2025; Liu et al., 2025b) are crucial for enhancing their reasoning abilities. Compared to approaches like Monte Carlo Tree Search (MCTS) (Wan et al., 2024) and Process Reward Modeling (Lightman et al., 2024; Wang et al., 2024; Zhao et al., 2025), Reinforcement Learning with Verifiable Rewards (RLVR) has emerged as a simple yet effective way to further improve the reasoning abilities of LLMs (Shao et al., 2024; Yu et al., 2025).

Recent studies have revealed that RL mainly improves reasoning by better integrating and organizing the model's existing abilities, such as reflection and planning, rather than directly changing the model's factual memory or basic skills (e.g., arithmetic) (Gandhi et al., 2025; Vassoyan et al., 2025; Li et al., 2025). Wang et al. (2025) also show that high-entropy tokens, which often act as logical connectors, are the main focus of RLVR adjustment, while low-entropy tokens mostly capture factual or domain knowledge. These findings together suggest an important principle: during RLVR training, the behavior of tokens tied to factual knowledge (low-entropy tokens) should change little compared to the base model, while tokens related to logical reasoning (high-entropy tokens) require stronger learning signals and greater exploration.

To account for different token types in RLVR, Wang et al. (2025); Cui et al. (2025b) use gradient masking to exclude low-entropy or high-covariance tokens from updates. Meanwhile, Yang et al. (2025b) introduces an asynchronous training method, updating different token types in separate gradient steps. While these methods agree on the need to treat low-entropy and high-entropy tokens differently based on metrics like entropy or token probability, we argue that these strategies have basic limitations: **tokens within a sentence and sentences within a response are closely related through semantic and syntactic dependencies, and require coordinated learning dynamics.** Completely stopping updates to low-entropy tokens breaks these dependencies, which in turn reduces the effective optimization of high-entropy reasoning tokens, as shown in Section 4.3.1 and 4.3.2.

To address these issues, we propose a synchronized, entropy-aware framework for differentiated token training. We use a response-level entropy criterion to group tokens into two types: (1) **knowledge-related tokens**, which mainly contain factual or domain-specific knowledge, and (2) **reasoning-related tokens**, which serve as logical connectors and guide step-by-step reasoning. Unlike earlier works that use masking or asynchronous updates, our method **synchronously** updates all tokens but applies **dual-token constraints** during training. Specifically, we set a higher clip threshold and weaker KL regularization for reasoning tokens to promote exploration and learning logical patterns, while for knowledge tokens, we use a lower clip threshold and stronger regularization to maintain factual accuracy.

We evaluate our approach on challenging mathematical reasoning and code generation benchmarks. Our experiments show significant performance improvements across different tasks. Compared to the standard DAPO algorithm (Yu et al., 2025), our dual-token constraints method achieves notable gains: +6.6 Pass@1 on `AIME24`, +5.2 on `AIME25`, +3.4 on `LiveCodeBench v5`, and +2.6 on `LiveCodeBench v6`. When compared with RL-trained models with the same base model, our approach achieves state-of-the-art performance on both mathematical and coding benchmarks. Beyond *pass@1* results, further analysis shows that our method also performs better on *pass@K* metrics, indicating a higher potential for reasoning abilities. In summary, our main contributions are:

- We propose an entropy-aware dual-token constraints framework that applies different clip and KL constraints in a synchronous update manner. This preserves knowledge on low-entropy tokens while improving reasoning ability on high-entropy tokens.

- Our empirical results show that the method achieves strong performance on challenging math and code reasoning tasks, outperforming DAPO and achieving better results than similarly sized models.

- We provide a systematic study of how KL weights and clip ranges affect the balance between preserving factual knowledge and encouraging reasoning exploration, showing how they can be used to control trade-offs in RL training.

## 2 PRELIMINARIES

### 2.1 GROUP RELATIVE POLICY OPTIMIZATION

Group Relative Policy Optimization (GRPO) (Shao et al., 2024) proposes an alternative to the value-based advantage estimation used in Proximal Policy Optimization (PPO) (Schulman et al., 2017). Instead of learning a value model, GRPO estimates advantages by sampling multiple rollouts per prompt. Specifically, for a given prompt $q$, GRPO generates a group of responses $\{o^1, o^2, \ldots, o^G\}$ and computes the corresponding rewards $\{R^1, R^2, \ldots, R^G\}$. The advantage is then calculated as:

$$\hat{A}_t^i = \frac{R^i - \text{mean}(\{R^i\}_{i=1}^G)}{\text{std}(\{R^i\}_{i=1}^G)}, \tag{1}$$

where both the mean and standard deviation are computed within the sampled group. The GRPO loss is computed as:

$$\mathcal{J}_{\text{GRPO}}(\theta) = \mathbb{E}_{q \sim \mathcal{D}, \{o^i\}_{i=1}^G \sim \pi_{\theta_{\text{old}}}(\cdot|q)}$$

$$\left[ \frac{1}{G} \sum_{i=1}^G \frac{1}{|o^i|} \sum_{t=1}^{|o^i|} \left( \min\left(r_t^i(\theta)\hat{A}_t^i, \text{clip}\left(r_t^i(\theta), 1-\varepsilon, 1+\varepsilon\right)\hat{A}_t^i\right) - \beta \mathbb{D}_{\text{KL}}(\pi_\theta \| \pi_{\text{ref}}) \right) \right], \tag{2}$$

where $r_t^i = \frac{\pi_\theta(o_t^i|q,o_{<t}^i)}{\pi_{\theta_{\text{old}}}(o_t^i|q,o_{<t}^i)}$ denotes the importance sampling ratio, and $\beta$ is a coefficient weighting the Kullback–Leibler (KL) divergence between the current policy $\pi_\theta$ and the reference policy $\pi_{\text{ref}}$.

### 2.2 DECOUPLE CLIP AND DYNAMIC SAMPLING POLICY OPTIMIZATION

Decouple Clip and Dynamic Sampling Policy Optimization (DAPO) (Yu et al., 2025) enhances GRPO by integrating four key techniques: Clip-Higher, Dynamic Sampling, Token-Level Policy Gradient Loss, and Overlong Reward Shaping. Similar to GRPO, DAPO samples multiple responses

per prompt and optimizes the policy using the following objective:

$$\mathcal{J}_{\text{DAPO}}(\theta) = \mathbb{E}_{(q,a)\sim\mathcal{D},\{o^i\}_{i=1}^G\sim\pi_{\theta_{\text{old}}}(\cdot|q)}$$

$$\left[\frac{1}{\sum_{i=1}^G|o^i|}\sum_{i=1}^G\sum_{t=1}^{|o^i|}\min\left(r_t^i(\theta)\hat{A}_t^i, \text{clip}\left(r_t^i(\theta), 1-\varepsilon_{\text{low}}, 1+\varepsilon_{\text{high}}\right)\hat{A}_t^i\right)\right] \quad (3)$$

$$\text{s.t.}\quad 0 < \left|\left\{i\in\{1,\ldots,G\}\mid\texttt{is\_equivalent}(o^i,a)\right\}\right| < G,$$

where $\varepsilon_{\text{low}}$ and $\varepsilon_{\text{high}}$ denote the lower and upper bounds of the clipping range.

## 3 METHOD

In this section, we introduce Archer, a novel RLVR approach with entropy-aware dual-token constraints. We begin by describing entropy-based method for identifying critical tokens (Section 3.1). Next, we discuss the limitations of prior methods in handling low-entropy tokens and motivate our approach for response-level entropy statistics (Section 3.1.1). We then analyze the necessity of joint training of high-entropy and low-entropy tokens (Section 3.2.1). Finally, we detail how Archer improves upon core constraints (clipping and KL) in previous RL algorithms by disentangling token-level optimization (Section 3.2.2).

### 3.1 CRITICAL TOKENS IDENTIFICATION VIA RESPONSE-LEVEL ENTROPY

Prior RL approaches like GRPO (Shao et al., 2024) and DAPO (Yu et al., 2025) typically adopt a uniform token-level optimization strength to all output tokens. This undifferentiated treatment fails to account for the distinct functional roles that different tokens play in the reasoning process (e.g., factual recall vs. logical decision points). Recent work shows that RL-driven improvements in LLM reasoning stem mainly from enhancing logical behaviors such as reflection and planning, which **integrate existing model capabilities**, rather than directly modifying the model's factual memory or primitive skills (Yue et al., 2025a; Wen et al., 2025). Thus, during RL training, tokens associated with *factual knowledge* or *base-level skills* should largely retain their original distributions, while tokens involved in *logical reasoning and decision-making* require stronger learning signals and targeted exploration. Identifying these critical reasoning tokens is therefore a crucial first step. To address this issue, a crucial first step is to identify critical reasoning tokens.

**Entropy-based Token Identification.** Recent work proposes entropy as an effective signal for identifying critical tokens, observing that high-entropy tokens frequently appear at logical transition points between reasoning segments (Wang et al., 2025). In contrast, low-entropy tokens typically complete ongoing statements or syntactic structures. This observation aligns with our hypothesis that entropy discriminates between reasoning-oriented and knowledge-oriented tokens. To empirically verify this, we analyze token entropy distributions of 1024 responses (each prompt 16 times) generated by `DeepSeek-R1-Distill-Qwen-1.5B` during training on mathematical tasks. Following Wang et al. (2025), we visualize the top-100 highest entropy tokens and the top-100 lowest entropy tokens and retain tokens that appear more than 100 times. The visualization in Figure 1 shows that high-entropy tokens are mainly reasoning-related tokens, while most low-entropy tokens are related to factual knowledge or the suffix part of a word. These findings are also validated by recent studies (Yang et al., 2025b; Cheng et al., 2025). In summary, token entropy serves as an effective metric to distinguish between reasoning-oriented and knowledge-oriented tokens.

### 3.1.1 RESPONSE-LEVEL ENTROPY STATISTICS

To distinguish token types, prior works compute token entropy quantiles or covariance statistics at the batch level (Wang et al., 2025; Cui et al., 2025b). However, we find this suboptimal due to substantial entropy variation across responses from different prompts, as shown in Figure 2. For instance, some prompts yield responses with average entropy far above/below the batch mean (Figure 2 (a)); even within a single prompt, entropy can vary across sampled responses significantly (Figure 2 (b)).

Therefore, batch-level statistics for token classification introduce a key drawback: if a response's overall entropy is low, even critical reasoning tokens may be misclassified as low-entropy, resulting in effective training. For example, using the 80th percentile as a threshold can result in only 4.34% of tokens being labeled as high-entropy in low-entropy responses. Conversely, for high-entropy responses, the proportion of high-entropy tokens may be abnormally inflated. To mitigate this, we adopt a **response-level** entropy statistics method for token classification, computing entropy quantiles

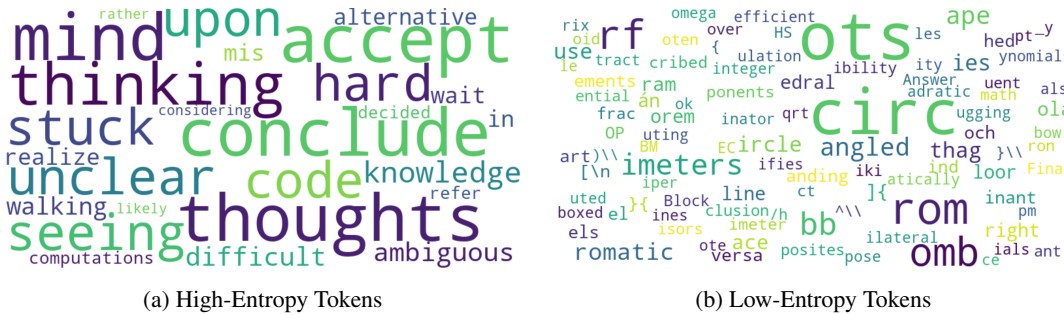

(a) High-Entropy Tokens  (b) Low-Entropy Tokens

Figure 1: Word cloud visualization of a batch of responses: (a) High-entropy tokens; (b) Low-entropy tokens.

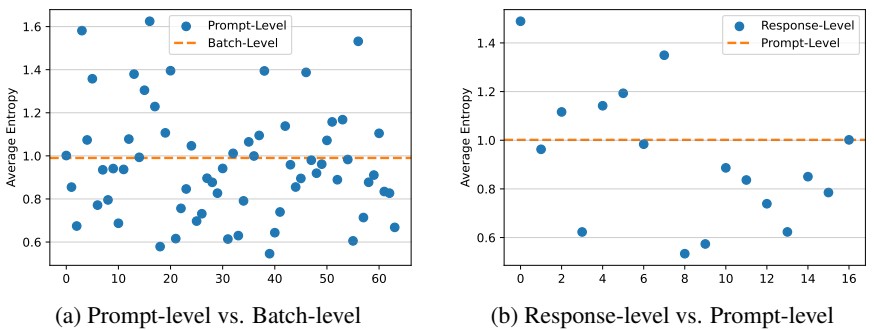

(a) Prompt-level vs. Batch-level  (b) Response-level vs. Prompt-level

Figure 2: Comparison of average entropy: (a) Prompt-level vs. batch-level across all prompts; (b) Response-level vs. prompt-level across all responses.

independently within each response. Given a batch of $N$ rollout responses, let $e_t^i$ be the entropy of token $t$ in response $o^i$. We compute the $\rho$-quantile of token entropy for each response as a threshold:

$$\tau_\rho^i = \text{Quantile}\left(\{e_t^i\}_{t=1}^{|o^i|}, \rho\right), \tag{4}$$

where $\rho \in (0, 1)$ denotes the quantile level (e.g., $\rho = 0.8$ corresponds to the 80th percentile).

### 3.2 TOKEN-LEVEL DISENTANGLED TRAINING

#### 3.2.1 PARTICIPATORY TRAINING OF LOW-ENTROPY TOKENS

To account for token type during RL training, recent works employ gradient masking for low-entropy tokens (Wang et al., 2025) or sequentially update different token types (Yang et al., 2025b). However, we argue that completely excluding or asynchronously updating low-entropy tokens is suboptimal. LLMs generate tokens sequentially, and the entropy of subsequent tokens is highly dependent on preceding content. As shown in Figure 3, high- and low-entropy tokens often interleave. The semantic and syntactic links among tokens and sentences require coordinated updates. If updates to low-entropy tokens are fully blocked or isolated, these dependencies are broken, which reduces effective learning for important high-entropy reasoning steps. To support this point, we conduct an ablation study in Section 4.3.2, where we change the clipping threshold for low-entropy tokens. The results show that as the clipping threshold becomes stricter (e.g., by setting the clip value of low-entropy tokens to 0), the model learns more slowly and its final performance drops.

#### 3.2.2 OUR METHOD

To address these issues, we propose a framework that performs synchronous updates while applying differentiated training constraints to different token types. Using response-level entropy as the criterion, we distinguish knowledge-type (low-entropy) from reasoning-type (high-entropy) tokens. Unlike prior works that adopt isolation strategies (e.g., gradient masking or asynchronous training), our method updates all tokens jointly, but applies different levels of training constraints to knowledge-type and reasoning-type tokens, respectively. Specifically, we target two core mechanisms in GRPO:

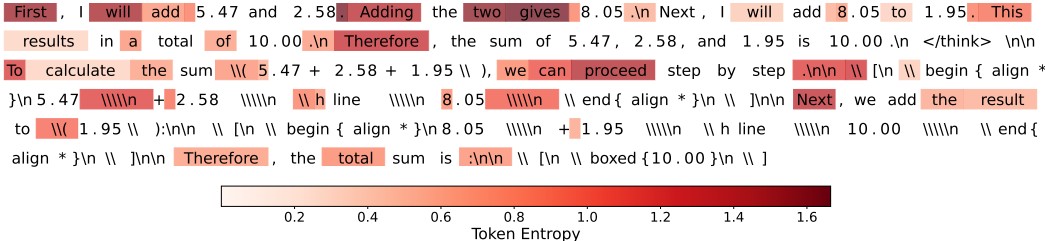

Figure 3: Visualization of high-entropy tokens (i.e., tokens with entropy larger than 80th percentile) within a response.

**Clipping Constraint.** To control the magnitude of policy updates at each step, we apply stricter clip ranges to knowledge-type (low-entropy) tokens to preserve the base model's capabilities and looser clip ranges to reasoning-type (high-entropy) tokens to encourage exploratory behavior. Given a batch of responses, we first compute the entropy quantile $\tau_\rho^i$ of token entropy within each response using equation 4. Based on the computed entropy threshold, we categorize tokens into different types and assign distinct clipping ranges to each type accordingly.

$$\varepsilon(e_t^i) = \begin{cases} \varepsilon^{\mathrm{r}} & \text{if } e_t^i \geq \tau_\rho^i, \\ \varepsilon^{\mathrm{k}} & \text{otherwise.} \end{cases} \tag{5}$$

**KL Constraint.** In RL training, the KL divergence penalty is commonly used to constrain the overall deviation of the trained policy from a reference policy (Shao et al., 2024). Although recent works (Yu et al., 2025; Liu et al., 2025c; Hu et al., 2025; Chu et al., 2025; Yue et al., 2025b; He et al., 2025) advocate removing the KL divergence penalty, ProRL (Liu et al., 2025a) argues that this typically holds for base models without extensive SFT and using the KL penalty is crucial for training stability. Our experimental results also confirm that fully removing the KL penalty leads to training collapse and degraded performance, as shown in Section 4.3.1. Moreover, applying uniform KL penalties across all tokens, including high-entropy ones, significantly slows learning and reduces final performance.

Therefore, we extend the conventional KL penalty by adapting it based on the functional type of each token. Specifically, we apply a stronger KL penalty (i.e., a larger KL weight) to knowledge-type tokens (low entropy) to preserve the base model's factual knowledge. In contrast, we apply a weaker KL penalty (i.e., a smaller KL weight) to reasoning-type tokens (high entropy), enabling greater flexibility in critical reasoning regions. The coefficients of KL constraints are as follows:

$$\beta(e_t^i) = \begin{cases} \beta^{\mathrm{r}}, & \text{if } e_t^i \geq \tau_\rho^i, \\ \beta^{\mathrm{k}}, & \text{otherwise.} \end{cases} \tag{6}$$

Finally, the overall objective of our algorithm is formulated as follows:

$$\mathcal{J}_{\mathrm{TDPO}}(\theta) = \mathbb{E}_{(q,a)\sim\mathcal{D},\{o^i\}_{i=1}^G \sim \pi_{\theta_{\mathrm{old}}}(\cdot|q)} \left[ \frac{1}{\sum_{i=1}^G |o^i|} \sum_{i=1}^G \sum_{t=1}^{|o^i|} \right.$$

$$\left. \left( \min\left( r_t^i(\theta)\hat{A}_t^i, \mathrm{clip}\left(r_t^i(\theta), 1 - {\color{red}\varepsilon(e_t^i)}, 1 + {\color{red}\varepsilon(e_t^i)}\right)\hat{A}_t^i\right) - {\color{red}\beta(e_t^i)}\mathbb{D}_{\mathrm{KL}}(\pi_\theta \| \pi_{\mathrm{ref}}) \right) \right] \tag{7}$$

$$\text{s.t.} \quad 0 < \left| \left\{ i \in \{1, \ldots, G\} \mid \mathtt{is\_equivalent}(o^i, a) \right\} \right| < G,$$

where differentiated clipping and KL constraints are denoted using red color. The full algorithm of Archer is shown in Algorithm 1.

## 4 EXPERIMENTS

### 4.1 SETUP

**Models and Baselines.** We adopt `DeepSeek-R1-Distill-Qwen-1.5B` as the base model, which is distilled from DeepSeek-R1 (DeepSeek-AI et al., 2025) using Qwen2.5-1.5B (Yang et al.,

2024) as the backbone and fine-tuned on 800k high-quality reasoning data. To ensure a fair comparison, we compare Archer against the following methods: (1) **Base Model**: The raw distilled model without further training. (2) **DAPO** (Yu et al., 2025): A RLVR algorithm that improves upon GRPO (Shao et al., 2024). (3) **DeepScaleR-1.5B** (Luo et al., 2025b): A 1.5B model trained on mathematical tasks with iterative context length expansion. (4) **DeepCoder-1.5B** (Luo et al., 2025a): A 1.5B model trained on code datasets, also utilizing context expansion strategies. (5) **FastCuRL-1.5B-V3** (Song et al., 2025): A strong 1.5B model with curriculum RL training. (6) **Nemotron-1.5B** (Liu et al., 2025a): Currently the best 1.5B reasoning model that RL-trained with `DeepSeek-R1-Distill-Qwen-1.5B` as the base model.

**Training Data.** For code domain, we construct a high-quality code training dataset from three publicly available sources: DeepCoder (Luo et al., 2025a), CodeContests (Li et al., 2022), and CodeForces (Penedo et al., 2025). Notably, CodeContests and CodeForces augment original problems with extensive test cases, which reduces false positives (i.e., incorrect solutions that pass test cases). Therefore, we prioritize these two datasets over DeepCoder in cases of duplication. After rigorous cleaning and filtering steps (detailed in Appendix C.1), we obtain a final corpus of **6,753** programming problems. For mathematics domain, we use datasets from DeepScaleR (Luo et al., 2025b), Skywork-OR1 (He et al., 2025), and DAPO (Yu et al., 2025). We merge these datasets and apply N-gram overlap removal to eliminate duplicates. After additional verification and filtering steps (see Appendix C.1), we derive a final mathematics training set of **51,800** problems.

**Evaluation and Metrics.** We conduct evaluation on both mathematical and coding benchmarks. For mathematics, we use six challenging datasets: AIME24 (MAA, 2024), AIME25 (MAA, 2025), AMC23 (MAA, 2023), MATH-500 (Lightman et al., 2024), Minerva Math (Lewkowycz et al., 2022), and OlympiadBench (He et al., 2024). For coding, we adopt the widely used LiveCodeBench v5 (2024.08.01-2025.02.01) and v6 (2025.02.01-2025.05.01) (Jain et al., 2025), which emphasize reasoning-intensive code generation. We use `vLLM` (Kwon et al., 2023) with temperature set to 0.8, `top_p` set to 1.0, and maximum output length set to 32,768 tokens for inference. Due to the high variance of the outputs from reasoning models, we report *avg@K* (pass@1 performance averaged over K outputs) and *pass@K* for each benchmark. For benchmarks with few samples (AIME24/25 and AMC23), we set a larger K=64. We use K=16 for LiveCodeBench v6, K=8 for LiveCodeBench v5 and Minerva, and K=4 for MATH-500 and OlympiadBench. To ensure accurate evaluation, we adopt the verification functions from both DeepScaleR and `Math-Verify`[1] for mathematics problems.

**Implementation Details.** We perform RL training using the `verl` framework (Sheng et al., 2024). For DAPO-based baselines, we use clipping thresholds of $\varepsilon_{\text{low}} = 0.2$ and $\varepsilon_{\text{high}} = 0.28$. KL penalty loss and entropy regularization loss are omitted from the loss function. During training, we sample 16 rollouts per prompt, with a temperature of 1.0 and a maximum response length of 32,768 tokens. The batch size is set to 64, the mini-batch size to 32, and the learning rate to $1 \times 10^{-6}$. For Archer, we set $\rho = 0.8$ following Wang et al. (2025). For clipping ranges and KL coefficients, we use $\varepsilon^{\text{r}} = 0.5$, $\varepsilon^{\text{k}} = 0.2$, $\beta^{\text{r}} = 0.0$, and $\beta^{\text{k}} = 0.001$. All experiments are conducted on 2 compute nodes, each equipped with $8 \times$ NVIDIA H800 80GB GPUs.

## 4.2 MAIN RESULTS

**Comparison with Base Model and DAPO.** The results in Table 1 and 2 show that our dual-token constraint training strategy leads to significant improvements on both mathematical and coding tasks. Compared to the original base model, the average accuracy increases by **18.1%** on AIME24 and **10.3%** on AIME25, resulting in an average gain of **12.3%**. On coding benchmarks, the accuracy rises by **12.7%** on LiveCodeBench v5 and **13.0%** on LiveCodeBench v6. When applying our method upon DAPO, the performance consistently exceeds that of DAPO across all benchmarks, with average gains of **5.6%** and **3.0%** for mathematical and coding tasks, respectively. These results demonstrate the effectiveness of our optimization approach.

**Comparison with SOTA Reasoning Models.** We also compare Archer with SOTA reasoning models trained with RL using `DeepSeek-R1-Distill-Qwen-1.5B` as the base model. For coding tasks, our approach outperforms all comparable models, including the programming-specialized **DeepCoder-1.5B** and the general-purpose **Nemotron-1.5B**. On mathematical reasoning, our model achieves the highest average accuracy, surpassing both math-specialized models (**DeepScaleR-1.5B**,

---

[1]https://github.com/huggingface/Math-Verify

Table 1: Evaluation results on mathematical benchmarks. The results of Archer are shaded and the highest values are **bolded**.

| Method | AIME24 | | AIME25 | | AMC23 | | MATH-500 | | Minerva | | Olympiad | | Avg. |
|---|---|---|---|---|---|---|---|---|---|---|---|---|---|
| | avg@64 | pass@64 | avg@64 | pass@64 | avg@64 | pass@64 | avg@4 | pass@4 | avg@8 | pass@8 | avg@4 | pass@4 | |
| DeepSeek-R1-1.5B | 30.6 | 80.0 | 23.5 | 63.3 | 70.7 | **100.0** | 83.6 | 92.4 | 27.6 | 48.2 | 44.6 | 59.4 | 46.8 |
| DAPO | 42.1 | 80.0 | 28.6 | 56.7 | 80.3 | 97.5 | 87.6 | **94.6** | 29.2 | 46.3 | 53.2 | 65.8 | 53.5 |
| DeepScaleR-1.5B | 42.0 | **83.3** | 29.0 | 63.3 | 81.3 | **100.0** | 87.7 | 93.6 | 30.3 | **51.1** | 50.7 | 61.0 | 53.5 |
| FastCuRL-1.5B-V3 | 48.1 | 80.0 | 32.7 | 60.0 | **86.4** | 95.0 | 89.8 | 94.0 | 33.6 | 50.0 | 55.3 | 64.3 | 57.7 |
| Nemotron-1.5B | 48.0 | 76.7 | 33.1 | 60.0 | 86.1 | 97.5 | 90.6 | 93.6 | 35.3 | 47.8 | 59.2 | 66.8 | 58.7 |
| Archer-Math-1.5B | **48.7** | **83.3** | **33.8** | **70.0** | 86.0 | 97.5 | **90.8** | 94.4 | **35.7** | **51.1** | **59.3** | **67.1** | **59.1** |

Table 2: Evaluation results on code benchmarks. The results of Archer are shaded and the highest values are **bolded**.

| Method | LCB v5 (2024.08.01-2025.02.01) | | LCB v6 (2025.02.01-2025.05.01) | | Avg. |
|---|---|---|---|---|---|
| | avg@8 | pass@8 | avg@16 | pass@16 | |
| DeepSeek-R1-1.5B | 16.7 | 29.0 | 17.2 | 34.4 | 17.0 |
| DAPO | 26.0 | 40.5 | 27.6 | 43.5 | 26.8 |
| DeepCoder-1.5B | 23.3 | 39.1 | 22.6 | 42.0 | 23.0 |
| Nemotron-1.5B | 26.1 | 35.5 | 29.5 | 42.8 | 27.8 |
| Archer-Code-1.5B | **29.4** | **43.7** | **30.2** | **45.8** | **29.8** |

**FastCuRL-1.5B-V3**) and **Nemotron-1.5B**. We report the training costs of Archer and these open-source reasoning models, including the number of training steps, stages, and GPU hours in Table 3. Notably, our model achieves the best results with only **single-stage** training and **fewer** GPU hours, without the complex multi-round training used by the other methods. In addition to improvements in **pass@1**, our model also shows advantages in **pass@K** metrics, which suggests stronger reasoning diversity and higher capability limits of our method.

Table 3: Computational efficiency comparison between Archer and the baselines.

| Method | Training Steps | Training Stages | GPU Hours |
|---|---|---|---|
| *Math RL* | | | |
| DeepScaleR-1.5B | 1750 | 3 | 3,800 A100 |
| FastCuRL-1.5B-V3 | 2620 | 5 | — |
| Nemotron-1.5B | 2500 | 8 | 16,000 H100 |
| Archer-Math-1.5B | 520 | 1 | 1,900 H800 |
| *Code RL* | | | |
| DeepCoder-1.5B | — | — | — |
| Nemotron-1.5B | 2500 | 8 | 16,000 H100 |
| Archer-Code-1.5B | 320 | 1 | 1,000 H800 |

## 4.3 ANALYSIS

### 4.3.1 IMPACT OF DIFFERENT KL WEIGHTS

In this part, we empirically study the impact of changing the KL penalty weight applied to low-entropy tokens during training. In addition to the default weight of 0.001 from earlier experiments, we conduct experiments with weights of 0.0 (i.e., no KL penalty) and 0.005. We calculate the average n-gram repetition ratio in generated outputs over training, which serves as a proxy for model collapse severity. The experimental results are shown in Table 4, with the corresponding training dynamics and repetition ratios visualized in Figure 5. Our results show that both the absence of KL regularization and an excessively high weight reduce performance. Specifically,

- **When KL weight = 0.0**: Figure 5(b) shows that model entropy drops rapidly, and there is a notable increase in repetition rate in Figure 5(c), indicating severe model collapse. Although performance on the LiveCodeBench v5 test set improves quickly at first, it soon levels off and shows limited gains in later stages. The final model not only underperforms the KL-regularized baseline but also falls below the standard DAPO method.

- **When KL weight = 0.005**: Entropy decreases more slowly, and repetition grows at a more gradual rate, better preserving the base model's characteristics. However, this setting slows down learning progress, resulting in smaller performance gains.

Table 4: Model performance on LiveCodeBench v5 with varying KL weights on low-entropy tokens.

| KL Weight | LiveCodeBench v5 (avg@8) |
|---|---|
| 0.0 | 26.6 |
| 0.001 | 29.4 |
| 0.005 | 26.2 |

In summary, both too little and too much KL regularization hurt the final model quality. Insufficient weighting accelerates learning but makes collapse more likely, which ends up reducing performance. In contrast, excessive weighting limits learning on low-entropy tokens and thus restricts the model's capabilities. These results **highlight the need for KL regularization on low-entropy tokens** to keep the model close to the base policy, which helps prevent collapse and retain key abilities. These observations further support our view that low-entropy tokens should be included in training, as masking them negatively affects overall learning.

### 4.3.2 IMPACT OF CLIP RANGES ON DIFFERENT TOKEN TYPES

We introduce different clip thresholds for different token types in equation 5. To investigate how the thresholds influence model performance, we vary the clip ranges for both high-entropy ($\varepsilon^r$) and low-entropy tokens ($\varepsilon^k$) and the results are shown in Table 5 and Figure 6, and Figure 7.

Table 5: Performance on LiveCodeBench v5 with different clip thresholds of low/high-entropy tokens.

| Low-Entropy Token Clip $\varepsilon^k$ | High-Entropy Clip $\varepsilon^r$ | LiveCodeBench v5 (avg@8) |
|---|---|---|
| *Varing Low-Entropy Token Clip* | | |
| 0.1 | 0.4 | 24.6 |
| 0.2 | 0.4 | 28.7 |
| 0.3 | 0.4 | 26.0 |
| *Varing High-Entropy Token Clip* | | |
| 0.2 | 0.2 | 27.7 |
| 0.2 | 0.4 | 28.7 |
| 0.2 | 0.5 | 29.4 |
| 0.2 | 0.6 | 26.0 |

**Different Low-Entropy Token Clip Thresholds.** As shown in Figure 6, we observe that increasing the clip threshold for low-entropy tokens produces effects similar to reducing their KL penalty weight: the model's entropy decreases more rapidly, which leads to faster learning and earlier performance improvements. However, this also causes the repetition ratio to rise more quickly, making the model more susceptible to overfitting or collapse, which harms final performance.

On the other hand, lowering the clip threshold for low-entropy tokens has effects similar to increasing their KL weight: improvements on LiveCodeBench v5 are slower and tend to converge a lower level. Interestingly, we observe an counterintuitive entropy dynamic during training. Instead of a consistently slow decline, as seen with higher KL weights, entropy initially drops sharply, then plateaus and remains stable.

These results indicate that adjusting the clip threshold for low-entropy tokens strongly affects both the training process and the final model performance. In contrast, the model is much less sensitive to changes in the clip threshold for high-entropy tokens.

**Different High-Entropy Token Clip Thresholds.** As illustrated in Figure 7, increasing the clip threshold for high-entropy tokens encourages more exploration in the model's reasoning. This leads to a slightly faster reduction in entropy during training and can improve the performance. However, these differences become more noticeable mainly in the later stages of training. In the early stages, training dynamics and LiveCodeBench v5 performance show little difference across various high-entropy clip values.

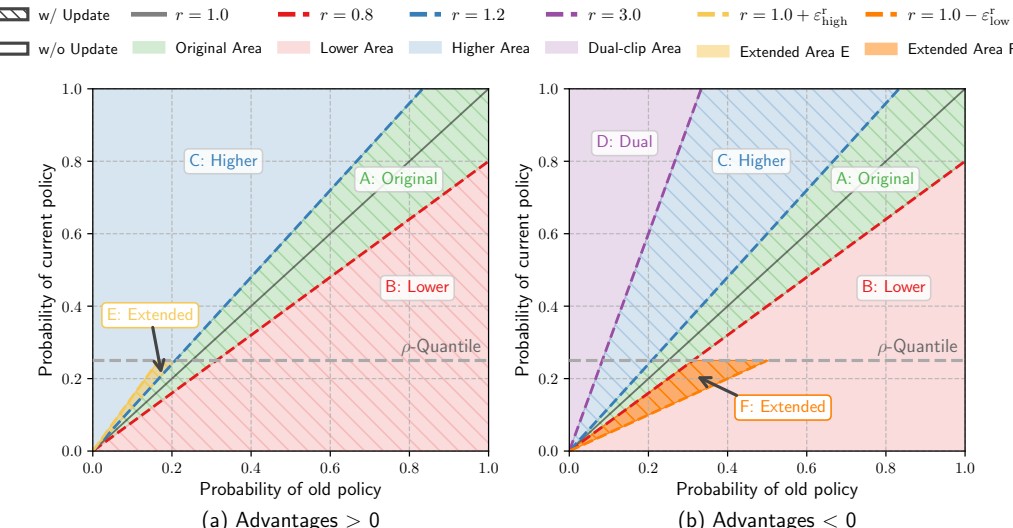

Figure 4: Visualization of PPO clip regions. The x-axis shows the sampled probability of a specific token $\pi_{\theta_{old}}$ during generation, and the y-axis shows the probability of the token under the current policy $\pi_\theta$. Region A represents the optimization area for original GRPO. Regions B and C represent areas below and above the clipping threshold, respectively. Region D is the area for dual-clip (Ye et al., 2020). (a) When advantages $> 0$, Archer optimizes region E. (b) When advantages $< 0$, Archer optimizes region F.

### 4.3.3 VISUALIZATION OF RL OPTIMIZATION REGIONS

To better clarify the mechanism of our method, we visualize the optimization regions produced by the GRPO loss for different token types in Figure 4. Each data point in the coordinate system represents the importance sampling ratio $r_t^i$ between the current and old policy probabilities. Figure 4(a) shows tokens with positive advantage values ($\hat{A}_t^i > 0$), while Figure 4(b) shows tokens with negative advantages ($\hat{A}_t^i < 0$). The colored regions mark the areas divided by the clipping thresholds. The shaded areas (Regions A, B for $\hat{A}_t^i > 0$ and Regions A, C for $\hat{A}_t^i < 0$) indicate where GRPO updates the model. Our method **extends the clipping boundaries for high-entropy tokens**, which are typically low-probability but are important for reasoning. As shown in Figure 4, Regions E and F correspond to the **newly extended optimization areas** introduced by Archer. Region E provides **additional reward signals** to high-entropy tokens when $\hat{A}_t^i > 0$, while Region F applies **stronger penalties** to high-entropy tokens when $\hat{A}_t^i < 0$. This design **increases the model's focus on learning reasoning-critical tokens**.

## 5 CONCLUSION

In this work, we propose an entropy-aware, synchronized training framework that updates all tokens simultaneously while applying different regularization and clipping strategies depending on the type of token. By encouraging exploration on reasoning-related tokens and preserving factual correctness for knowledge-related tokens, our method balances the goals of keeping factual accuracy and improving logical reasoning. Extensive experiments on mathematical and code reasoning benchmarks show that our approach improves over the base model and outperforms existing SOTA models. These results indicate that coordinating the learning processes of different token types through entropy-aware constraints improves the reasoning abilities of LLMs. We believe this work highlights the interaction between factual knowledge and reasoning processes during RL training of LLMs, and suggests future research directions for fine-grained, token-level optimization strategies that respect the inherent structural dependencies in natural language generation.

## REPRODUCIBILITY STATEMENT

We have discuss the implementation details in Section 4 and Appendix C. The code will be open-sourced after paper acceptance.

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

## A  FULL ALGORITHM

---
**Algorithm 1** Archer

---
**Input:** Base model $\pi_{\text{base}}$, prompt dataset $\mathcal{D}$, quantile level $\rho$, clipping thresholds $\varepsilon^{\text{r}}, \varepsilon^{\text{k}}$, KL coefficients $\beta^{\text{r}}, \beta^{\text{k}}$
  1: Initialize policy model $\pi_\theta \leftarrow \pi_{\text{base}}$ and reference model $\pi_{\text{ref}} \leftarrow \pi_{\text{base}}$
  2: **for** step $= 1, 2, \ldots, T$ **do**
  3:     Sample a batch of prompts $\mathcal{D}_b$ from $\mathcal{D}$
  4:     Generate responses $\{o^i\}_{i=1}^G$ for each prompt $q$ in the batch
  5:     **for** each response $|o^i|$ **do**
  6:         Compute the $\rho$-quantile of token entropy $\tau_\rho^i$ with equation 4
  7:         Compute clipping thresholds and coefficients of KL penalty with equation 5 and equation 6
  8:     **end for**
  9:     Update the policy model $\pi_\theta$ using equation 7
 10: **end for**

---

## B  RELATED WORK

### B.1  REINFORCEMENT LEARNING FOR LARGE LANGUAGE MODELS

Previous works have shown that RL, particularly Reinforcement Learning from Human Feedback (RLHF) (Christiano et al., 2017; Liu et al., 2022), is an effective tool for aligning LLMs with human preferences (Ouyang et al., 2022; Bai et al., 2022). With the recent success of scaling RL in LLMs (OpenAI, 2024a; DeepSeek-AI et al., 2025; Kimi Team et al., 2025), RLVR has emerged as an effective method to improve the reasoning ability of LLMs using rule-based rewards. However, approaches like GRPO (Shao et al., 2024) and its extensions (Yu et al., 2025; Liu et al., 2025c; Chu et al., 2025; Yue et al., 2025b; He et al., 2025) rely on response-level learning signals, which uniformly assign the same advantage value to all tokens within a response. This uniform treatment overlooks the distinct roles tokens play during reasoning (e.g., factual recall vs. logical inference), potentially leading to suboptimal learning at critical reasoning steps and limiting overall performance gains. Although process-based RL (Kazemnejad et al., 2025; Cui et al., 2025a; Zha et al., 2025) and unsupervised RL (Agarwal et al., 2025; Cheng et al., 2025) provide fine-grained rewards for RL optimization, they still lack consideration for the functions of different tokens.

### B.2  CRITICAL TOKEN ANALYSIS IN RL FOR REASONING

Several recent studies have provided token-level analyses of RLVR training (Yang et al., 2025b; Cui et al., 2025b; Wang et al., 2025; Cheng et al., 2025). Yang et al. (2025b) observe that low-probability tokens, often exhibiting high entropy, dominate the RL updates and the update of high-probability tokens are suppressed. Cui et al. (2025b) show that changes in policy entropy are linked to the covariance between action probabilities and advantages. Wang et al. (2025) identify high-entropy tokens, referred to as "forking tokens", as logical connectors. Cheng et al. (2025) further associate high-entropy tokens with reasoning-related behaviors, such as logical transitions and self-reflection. Unlike prior works that either completely isolate low-entropy tokens (Wang et al., 2025) or high-covariance tokens (Wang et al., 2025; Cui et al., 2025b), or train them separately (Yang et al., 2025b), our approach employs joint training. While we similarly utilize entropy to distinguish between logic-oriented and knowledge-oriented tokens, we avoid direct filtering or separation. Instead, we apply differentiated training constraints, enabling us to preserve the capabilities of the base model while simultaneously encouraging more effective exploration during training.

## C  EXPERIMENTAL DETAILS

### C.1  DATASET

#### C.1.1  CODE DOMAIN

**Data Sources and Integration.** The code dataset is compiled from three publicly available sources: DeepCoder, CodeContests, and CodeForces. Notably, CodeContests and CodeForces extend their original problem sets with a larger number of test cases, improving the reliability of evaluation and reducing the incidence of false positives—i.e., incorrect code that inadvertently passes tests. As such,

these two datasets are prioritized. In cases of duplication with DeepCoder, we retain the entries from either CodeContests or CodeForces.

**Data Cleaning and Filtering Pipeline.**   We apply a rigorous multi-stage cleaning and selection process to ensure dataset quality:

1. **Test Case Preprocessing:** We remove illustrative test cases embedded in problem descriptions and discard problems with fewer than five test cases, which are more susceptible to false positives.

2. **Model Validation and Difficulty Filtering:** Each problem is evaluated using 8-sample generation with a strong language model (`Qwen3-30B-A3B` (Yang et al., 2025a)). We exclude problems for which all samples fail verification, filtering out flawed questions (e.g., with invalid test cases), overly long I/O problems beyond the verifier's capacity, or those that are excessively difficult—even for strong models. This reduces potential false negatives.

3. **Problem Deduplication:** We perform N-gram-level deduplication to eliminate duplicate questions within the training corpus.

4. **Test Set Contamination Prevention:** To prevent data leakage, we remove any overlapping problems by conducting N-gram-level deduplication against the evaluation set of LiveCodeBench v5.

5. **Sampling Stability Filtering:** Using a warm-start model (`DeepSeek-R1-Distill-Qwen-1.5B`), we generate 8 additional samples per problem. We remove problems where all generations are either completely correct or completely incorrect, thereby ensuring sufficient learning signal and gradient diversity.

**Data Standardization.**   All retained code problems are reformatted into either *function-call* or *stdin/stdout* formats, enabling consistent and automated validation via a code verifier.

**Final Dataset.**   Following the aforementioned pipeline, we construct a high-quality code training dataset consisting of **6,753** problems.

### C.1.2   MATHEMATICS DOMAIN

**Data Sources and Integration.**   For the mathematics domain, we leverage existing curated datasets rather than raw symbolic corpora such as NuminaMath (LI et al., 2024). Specifically, we integrate three high-quality, verifiable datasets: DeepScaleR, Skywork-OR1, and DAPO. The datasets are merged and deduplicated using N-gram overlap removal to eliminate redundancy.

**Data Cleaning and Filtering Pipeline.**

1. **Model Validation and Filtering:** Each math problem undergoes 8-sample generation using the `Qwen3-30B-A3B` model, followed by verification using a mathematical logic verifier. Problems for which all samples fail are excluded to remove noise, overly complex items, or verification bottlenecks that might cause false negatives.

2. **Sampling Stability Filtering:** We repeat the 8-sample generation process using a warm-start model (`DeepSeek-R1-Distill-Qwen-1.5B`) and discard problems with homogeneous sampling outcomes (i.e., all correct or all incorrect).

3. **Test Set Contamination Prevention:** To avoid contamination of evaluation benchmarks, we perform N-gram deduplication against the AMC competition datasets (`AIME24` and `AIME25`), ensuring zero overlap.

**Final Dataset.**   After rigorous verification and filtering, we obtain a final mathematics training corpus comprising approximately **51,800** high-quality problems suitable for reinforcement learning.

## D   ADDITIONAL EXPERIMENTAL RESULTS

### D.1   IMPACT OF CLIP RANGES ON HIGH-ENTROPY TOKENS

### D.2   MUTUAL ENHANCEMENT BETWEEN MATH RL AND CODE RL

Figure 8 shows results on AIME24, AIME25, and LiveCodeBench v5, comparing RL applied to math tasks (math RL) and code tasks (code RL). We observe that RL training in either domain

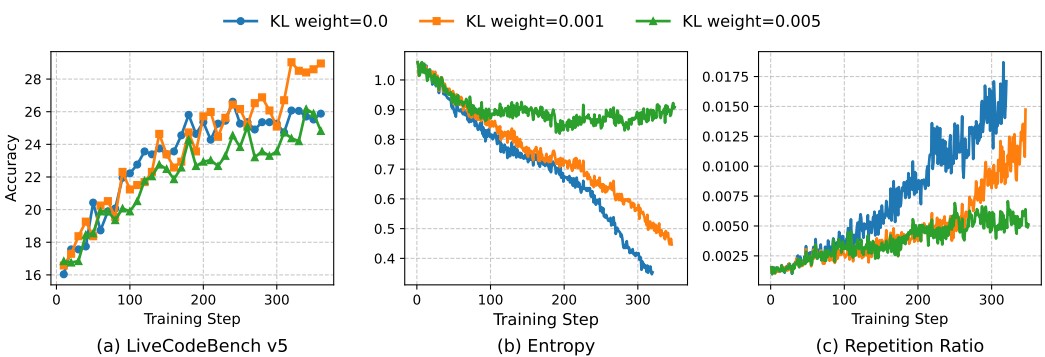

Figure 5: Effects of varying KL weights on (a) model performance on LiveCodeBench v5, (b) model entropy, and (c) repetition ratio.

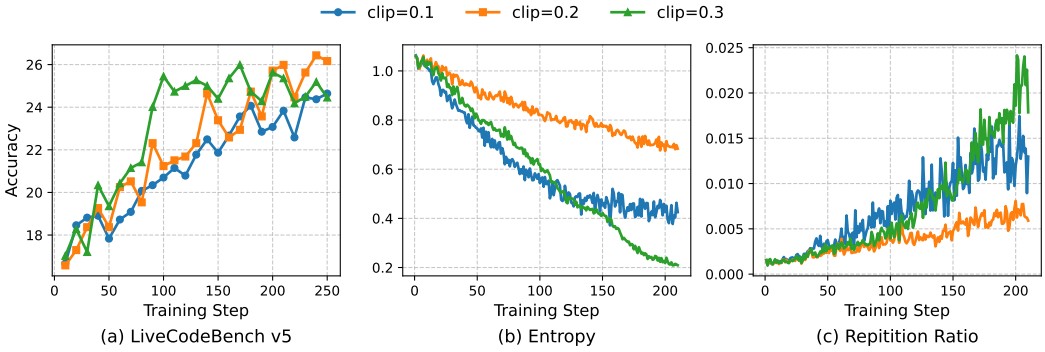

Figure 6: Effects of varying the clip threshold of low-entropy tokens on (a) model performance on LiveCodeBench v5, (b) model entropy, and (c) repetition ratio.

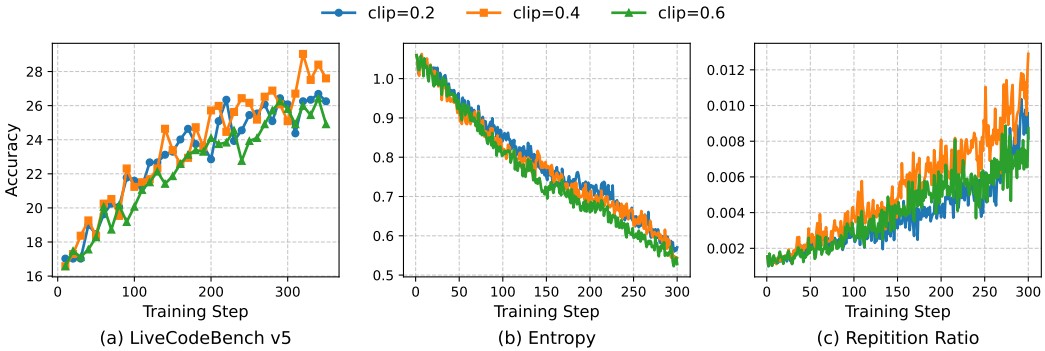

Figure 7: Effects of varying clip value on high-entropy tokens on (a) model performance on Live-CodeBench v5, (b) model entropy, and (c) repetition ratio.

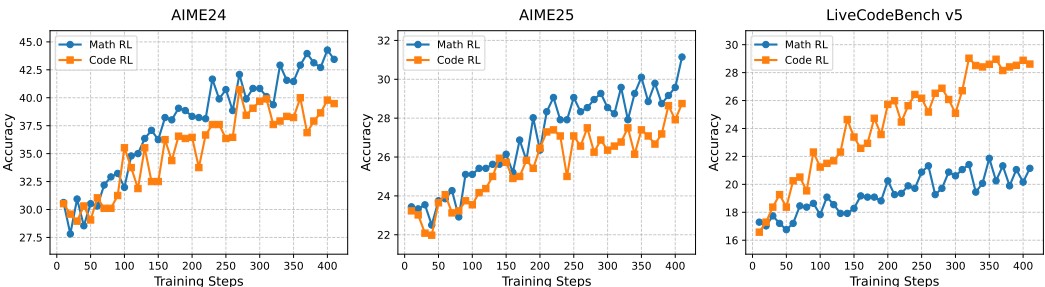

Figure 8: Model performance on AIME24, AIME25, and LiveCodeBench v5 of math RL and code RL.

leads to significant performance improvements not only in-domain but also on out-of-domain (OOD) benchmarks.

To analyze the source of these cross-domain improvements, we evaluate the base model and its math/code RL variants on OOD benchmarks (LiveCodeBench v5 and AIME24/25), measuring problem-level accuracy across all tasks. Unlike AceReason-Nemotron (Chen et al., 2025), which attributes the benefits of math RL on code tasks primarily to the presence of math-related subdomains (e.g., Algebra, Counting, Combinatorics), our results suggest a different explanation: performance improvements correlate more strongly with the intrinsic difficulty of the problems rather than their topical categories. Specifically, problems where the base model already achieves relatively high accuracy tend to benefit most from RL training, as shown in Figure 9 and Figure 10.

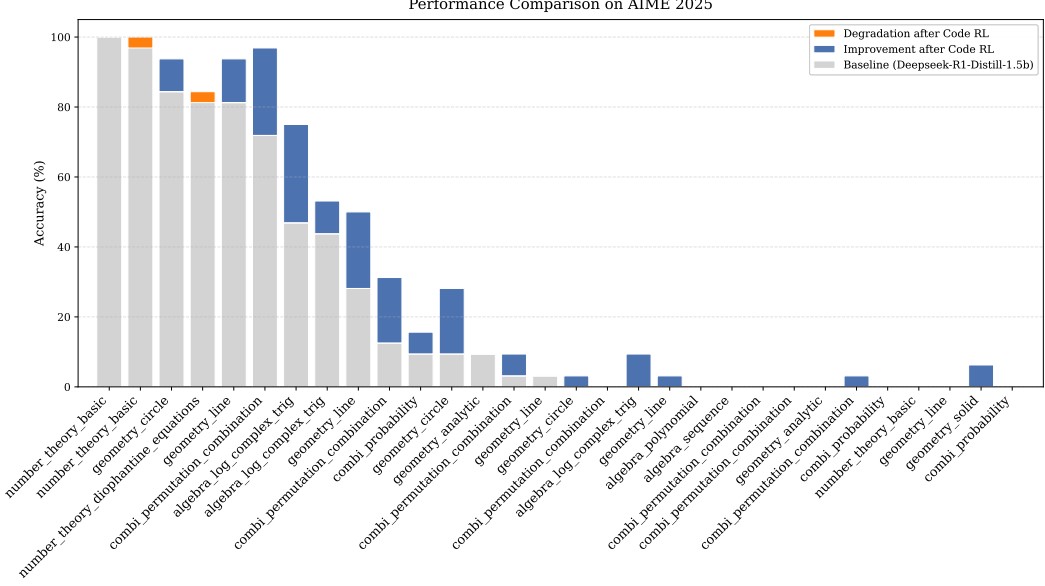

Figure 9: Problem-level accuracy comparison between the base model and RL-trained model.

A closer analysis of the problems with notable improvement in Figure 9 shows that RL training **does not introduce fundamentally new knowledge** beyond what is already present in the base model's outputs. This observation applies to both less challenging problems (where the base model already performs well) and more challenging ones. Instead, the improvements **mainly result from enhanced reasoning capabilities**. We identify three main areas of improvement:

- **Enhanced Structural Organization**: Responses demonstrate a clearer logical flow and improved structural coherence.

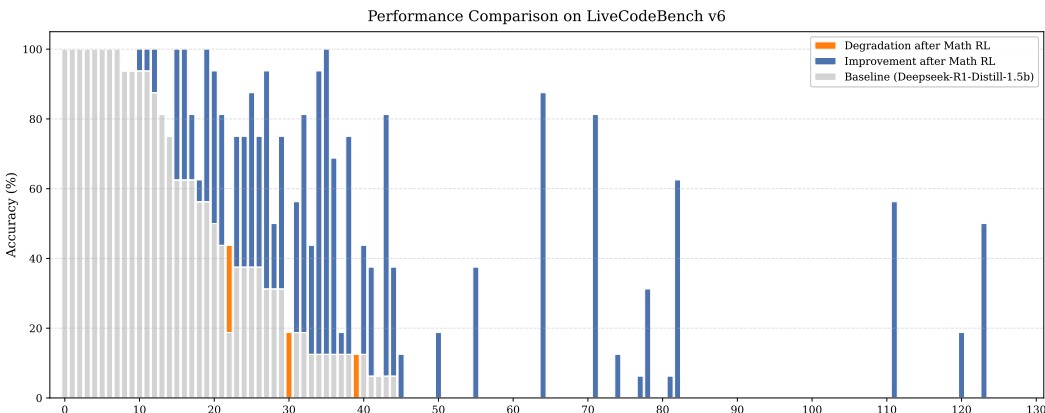

Figure 10: Problem-level accuracy comparison on LiveCodeBench v6 between the base model and Math RL trained model.

- **Increased Attention to Details**: Models are more careful with edge cases and boundary conditions. This effect is especially clear in the Code-RL model, likely because boundary handling is important in programming tasks.

- **Improved Contextual Consistency**: RL-trained models are more accurate at integrating and summarizing previous reasoning steps. In contrast, the base model sometimes produces final answers based on incorrect intermediate reasoning even if some steps are correct, which leads to inconsistencies.

These findings further support our main claim: the main way RL improves model capability **is not by changing stored knowledge or basic skills** (such as arithmetic), **but by better integrating and optimizing existing abilities** through structured logical behavior such as reflection and planning. At the same time, this provides empirical support for the effectiveness of our proposed dual-token constraint training strategy.

## E  THE USE OF LARGE LANGUAGE MODELS

We only use LLMs to polish some sections instead of directly using them for writing all words in this paper.

