# OpenReview forum: "Stabilizing Knowledge, Promoting Reasoning: Dual-Token Constraints for RLVR"
_ICLR.cc/2026/Conference — Submitted to ICLR 2026_

### Official Review · Reviewer_XjH3 · 2025-10-31

**Soundness:** 3
**Presentation:** 3
**Contribution:** 3
**Rating:** 8
**Confidence:** 4

**Summary:**

The paper proposes an entropy-aware RLVR method that splits tokens within each response into low-entropy knowledge tokens and high-entropy reasoning tokens using a response-level quantile. It updates both categories synchronously but applies different constraints: tighter clipping and stronger KL regularization for knowledge tokens to preserve factual style, and looser clipping with weaker KL for reasoning tokens to encourage exploration. The method integrates into GRPO or DAPO by making clipping ranges and KL weights token wise. On math and code benchmarks the approach improves accuracy and training stability over DAPO with single stage training and modest compute.

**Strengths:**

1. Clear motivation that probability mass concentration and uniform signals across heterogeneous tokens can either destabilize knowledge or suppress reasoning.
2. Response-level quantile avoids cross-prompt entropy mismatch and provides an adaptive way to select salient tokens without global thresholds.
3. Simple to implement in existing RLVR pipelines by turning clipping and KL into token wise functions.
4. Competitive results on math and code

**Weaknesses:**

1. Missing head-to-head baselines against other token balancing methods, especially High Entropy Minority Tokens and Low Probability Dominance with Advantage Reweighting or Lopti. Indirect ablations that freeze low-entropy tokens are not a substitute for these established recipes.
2. Limited analysis of sensitivity to the response-level quantile and to the pair of clipping and KL hyperparameters. The stability gains may depend on narrow settings.

**Questions:**

1. Have the authors considered providing a direct comparison to (or against) token balancing methods, such as those proposed in *Do Not Let Low-Probability Tokens Over-Dominate in RL for LLMs* or *Beyond the 80/20 Rule: High-Entropy Minority Tokens Drive Effective Reinforcement Learning for LLM Reasoning*？
2. Would a continuous weighting over tokens based on entropy or confidence outperform the binary split, and have the authors tried more than two levels within a response？

---

> ### Author Response · Authors · 2025-12-03
>
> We thank reviewer XjH3 for the positive support and insightful comments. We provide our point-wise responses below.
>
> ---
>
> ### **W1: "Missing head-to-head baselines against other token balancing methods, especially High Entropy Minority Tokens and Low Probability Dominance with Advantage Reweighting or Lopti. Indirect ablations that freeze low-entropy tokens are not a substitute for these established recipes."**
>
> **A**: Thank you for the suggestion. Due to compute limits, we have not run direct comparisons with the latest token balancing methods. We agree that these comparisons are important and will include them in the final version under controlled settings.
>
> ---
>
> ### **W2: "Limited analysis of sensitivity to the response-level quantile and to the pair of clipping and KL hyperparameters. The stability gains may depend on narrow settings."**
>
> **A**:
>
> **(1) Quantile**: We use the 80th percentile following prior work [1], which shows that high-entropy minority tokens are critical for reasoning.
>
> **(2) Clipping and KL hyperparameters**: We have conducted extensive ablation studies on the clipping and KL hyperparameters in Table 4 and 5 and Appendix D. The results indicate that while performance is relatively sensitive to these hyperparameters, there exists a **reasonable and explainable range** where the model performs well. For example, the commonly used clipping hyperparameter is $\epsilon=0.2$. To facilitate the learning of high-entropy tokens, we increase the clipping range to $\epsilon^r=0.3$ or $0.4$. For low-entropy tokens, we apply a stronger KL regularization with $\beta^k$ ranging from $0.1$ to $0.3$.
> We will include additional analysis in the final version to further clarify this point.
>
> ---
>
> ### **Q1: "Have the authors considered providing a direct comparison to (or against) token balancing methods, such as those proposed in Do Not Let Low-Probability Tokens Over-Dominate in RL for LLMs or Beyond the 80/20 Rule: High-Entropy Minority Tokens Drive Effective Reinforcement Learning for LLM Reasoning？"**
>
> **A**: We agree that comparison with these methods is important. We will include head-to-head baselines in the final version.
>
> ---
>
> ### **Q2: "Would a continuous weighting over tokens based on entropy or confidence outperform the binary split, and have the authors tried more than two levels within a response？"**
>
> **A**: Thanks for this insightful question. In this work, we focus on a binary classification of tokens based on entropy following [1], as it provides a simple yet effective way to differentiate between reasoning and knowledge tokens. However, we acknowledge that a continuous weighting scheme or a multi-level classification could potentially capture more nuanced information about token importance. We have not yet explored these alternatives in our current experiments, but we agree that they are promising directions for future research.
>
> **References**
>
> [1] Beyond the 80/20 Rule: High-Entropy Minority Tokens Drive Effective Reinforcement Learning for LLM Reasoning. NeurIPS 2025.

---

### Official Review · Reviewer_Hrnn · 2025-11-01

**Soundness:** 2
**Presentation:** 2
**Contribution:** 2
**Rating:** 4
**Confidence:** 3

**Summary:**

As proposed in the article, tokens in RLVR can be divided into two categories based on their entropy values: low-entropy knowledge-related tokens and high-entropy reasoning-related tokens. The adoption of varied CLIP ranges and KL weights for these two types of tokens yields enhanced experimental results, effectively reconciling the relationship between stability and exploration.

**Strengths:**

1. The proposed method is simple and easy to apply and optimize within existing frameworks.
2. The paper is well-written and easy to follow, with plenty of visualizations to help convey the key findings.

**Weaknesses:**

1. Please refer to the "Questions" section for details.

**Questions:**

1. Experimental evidence and concise discussions are provided, yet a theoretical justification for the two proposed improvements remains absent; it would be valuable to understand why they are effective.
2. The authors systematically vary the hyper-parameters associated with each improvement, but a rigorous ablation study that entirely ablates each component appears to be missing.
3. Is the method compatible with PPO, and has its applicability been analyzed?

---

> ### Author Response · Authors · 2025-12-03
>
> We thank reviewer Hrnn for the insightful comments. We provide our point-wise responses below.
>
> ---
>
> ### **Q1: "Experimental evidence and concise discussions are provided, yet a theoretical justification for the two proposed improvements remains absent; it would be valuable to understand why they are effective."**
>
> **A**: Thanks for your question. The entropy-based token classification is motivated by the observation in [1] that high-entropy tokens are more uncertain and likely to require reasoning, while low-entropy tokens are more predictable and likely to represent knowledge. By applying different constraints to these two types of tokens during policy optimization, we can better balance exploration and exploitation in RLVR. The clipping mechanism for high-entropy tokens encourages exploration and facilitates reasoning, while the KL regularization for low-entropy tokens helps maintain knowledge retention. This approach aligns with the principles of RL, where exploration is crucial for discovering new strategies, and exploitation is necessary for leveraging existing knowledge. We will include a more detailed explanation in the final version.
>
> ---
>
> ### **Q2: "The authors systematically vary the hyper-parameters associated with each improvement, but a rigorous ablation study that entirely ablates each component appears to be missing."**
>
> **A**: Thank you for the suggestion. We have indeed provided this ablation study in Table 4 and 5. For example, $\epsilon^k=0.2$ means removing the addtional control of low-entropy tokens, and $\epsilon^r=0.2$ means removing the additional control of high-entropy tokens. The results show that removing either component leads to performance degradation, demonstrating the effectiveness of both components.
>
> ---
>
> ### **Q3: "Is the method compatible with PPO, and has its applicability been analyzed?"**
>
> **A**: Yes, the method is compatible with PPO. Archer only modifies token-level clip and KL constraints, which can be applied to PPO. We use GRPO because recent SOTA models [1, 2] prefer critic-free methods. We will add analysis on PPO in future work.
>
> **References**
>
> [1] Beyond the 80/20 Rule: High-Entropy Minority Tokens Drive Effective Reinforcement Learning for LLM Reasoning. NeurIPS 2025.
>
> [2] ProRL: Prolonged Reinforcement Learning Expands Reasoning Boundaries in Large Language Models. NeurIPS 2025.

---

### Official Review · Reviewer_uydx · 2025-11-01

**Soundness:** 3
**Presentation:** 2
**Contribution:** 3
**Rating:** 6
**Confidence:** 3

**Summary:**

This paper primarily propsed a method to balance the stability of factual knowledge with the enhancement of reasoning abilities when using Reinforcement Learning with Verifiable Rewards (RLVR) to train Large Language Models (LLMs). The research finds that traditional RLVR algorithms apply a uniform training signal to all tokens, which ignores their different roles: low-entropy tokens are often associated with factual knowledge and should remain stable, while high-entropy tokens are typically linked to logical reasoning and require more exploration. To solve this, the paper proposes the Archer framework to use synchronous updates combined with dual-token constraints.
Experimental results show that Archer significantly outperforms baselines like DAPO on mathematical reasoning and code generation benchmarks, achieving state-of-the-art (SOTA) performance among models of comparable size.

**Strengths:**

- Significant Performance Gains. The method achieves outstanding results on several challenging math and code benchmarks. Compared to the baseline DAPO algorithm, Archer demonstrates significant gains, such as +6.6 Pass@1 on AIME24, and achieves SOTA performance among similarly sized models.
- Higher Training Efficiency: The paper reports that unlike other SOTA models that rely on complex multi-stage or multi-round training, Archer achieves its best average accuracy with only single-stage training and fewer GPU hours.
- Thorough Empirical Analysis on a specific model. The paper strongly supports its core claims with detailed ablation studies.

**Weaknesses:**

- Introduction of Hyperparameters. The method introduces several hyperparameters that require careful tuning, including the clipping ranges ($\epsilon^{k}$, $\epsilon^{r}$) and KL weights ($\beta^{k}$, $\beta^{r}$) for both token types. The ablation studies show that model performance is quite sensitive to these values (especially $\beta^{k}$), which may increase the difficulty of reproducing the best results on different tasks or models.
- Limited Generalizability of Model and Task. All experiments are conducted on a specific 1.5B parameter base model (DeepSeek-R1-Distill-Qwen-1.5B). Furthermore, the evaluations are focused on math and code, which are highly structured reasoning domains. It is unclear if this method would be equally effective on much larger models (e.g., 100B+) or different types of tasks (e.g., common-sense reasoning, creative writing).
- Oversimplified Token Classification. Archer relies on a binary classification based on entropy to distinguish "knowledge" from "reasoning" tokens. This binary split might be too simplistic, as a token's function in reality could be complex, multidimensional, or fall on a spectrum, rather than always being clearly defined by the single metric of entropy.

**Questions:**

The paper chose the 80th percentile of response-level entropy as the threshold for distinguishing high- and low-entropy tokens. How was this specific threshold determined, and how sensitive is the model's performance to this quantile value?

---

> ### Author Response · Authors · 2025-12-03
>
> We thank reviewer uydx for the positive support and insightful comments. We provide our point-wise responses below.
>
> ---
>
> ### **W1: "Introduction of Hyperparameters. The method introduces several hyperparameters that require careful tuning, including the clipping ranges ($\epsilon^k$, $\epsilon^r$) and KL weights ($\beta^k$, $\beta^r$) for both token types. The ablation studies show that model performance is quite sensitive to these values (especially $\beta^k$), which may increase the difficulty of reproducing the best results on different tasks or models."**
>
> **A**: We agree that the hyperparameters require careful tuning. Our ablation studies show that $\beta^k$ has the largest influence because low-entropy tokens are closely tied to factual knowledge. Although sensitivity exists, we also observe a stable range where the method performs well. We will provide clearer guidance and a practical selection rule in the final version.
>
> ---
>
> ### **W2: "Limited Generalizability of Model and Task. All experiments are conducted on a specific 1.5B parameter base model (DeepSeek-R1-Distill-Qwen-1.5B). Furthermore, the evaluations are focused on math and code, which are highly structured reasoning domains. It is unclear if this method would be equally effective on much larger models (e.g., 100B+) or different types of tasks (e.g., common-sense reasoning, creative writing)."**
>
> **A**: Our evaluation setup follows previous work [1, 2, 3], which also focuses on math and code. We agree that it is important to test the method on larger models or more diverse tasks. We will include results on 7B models in the final version. Experiments on 100B models are not feasible for us at the moment.
>
> ---
>
> ### **W3: "Oversimplified Token Classification. Archer relies on a binary classification based on entropy to distinguish "knowledge" from "reasoning" tokens. This binary split might be too simplistic, as a token's function in reality could be complex, multidimensional, or fall on a spectrum, rather than always being clearly defined by the single metric of entropy."**
>
> **A**: We agree that token roles can be complex. Our binary split is a simple and effective choice supported by prior work [1]. Our experiments also show that using different constraints for the two types improves optimization. We will explore continuous weighting schemes or multi-level splits in future work.
>
> ---
>
> ### **Q1: "The paper chose the 80th percentile of response-level entropy as the threshold for distinguishing high- and low-entropy tokens. How was this specific threshold determined, and how sensitive is the model's performance to this quantile value?"**
>
> **A**: We select the 80th percentile based on [1], which shows the importance of high-entropy minority tokens. Our response-level threshold is more adaptive than the batch-level threshold in [1].
>
> **References**
>
> [1] Beyond the 80/20 Rule: High-Entropy Minority Tokens Drive Effective Reinforcement Learning for LLM Reasoning. NeurIPS 2025.
>
> [2] ProRL: Prolonged Reinforcement Learning Expands Reasoning Boundaries in Large Language Models. NeurIPS 2025.
>
> [3] DeepScaleR: Surpassing O1-Preview with a 1.5B Model by Scaling RL. Notion Blog, 2025.

---

### Official Review · Reviewer_2PyW · 2025-11-03

**Soundness:** 3
**Presentation:** 3
**Contribution:** 3
**Rating:** 6
**Confidence:** 4

**Summary:**

The proposed method, Archer, introduces a simple training recipe for RL with verifiable rewards that treats tokens differently based on response‑level entropy while keeping synchronous updates: high‑entropy “reasoning” tokens receive looser clipping and weaker KL to encourage exploration, and low‑entropy “knowledge” tokens receive tighter clipping and stronger KL to preserve factual recall. The method drops cleanly into GRPO/DAPO‑style trainers and shows consistent gains on both math (AIME24/25, AMC23, MATH‑500, Minerva, OlympiadBench) and code (LiveCodeBench v5/v6), with ablations that illuminate why keeping some KL on low‑entropy tokens and carefully setting clip ranges matter for stability and final quality.

**Strengths:**

1. This proposed method adds a clear, implementation‑friendly mechanism—token‑typed constraints from response‑level entropy—that integrates seamlessly with existing GRPO/DAPO setups. The decision to keep synchronous updates while relaxing constraints on reasoning tokens is well motivated, and the visual analysis of optimization regions and token interleaving makes the intuition concrete.

2. The paper demonstrates consistent improvements across math and code benchmarks, suggesting the approach benefits reasoning structure rather than a single task or dataset. Ablations on KL weights and clip ranges are thoughtfully designed, revealing collapse‑vs‑stability trade‑offs and guiding practitioners toward robust settings

**Weaknesses:**

1. All results rely on a single 1.5B base (DeepSeek‑R1‑Distill‑Qwen‑1.5B); adding a second backbone or a larger‑scale model would strengthen generality. Further, The entropy quantile (ρ) is fixed without a sensitivity study, and the work would benefit from either a small sweep or an adaptive rule of thumb.

2. No direct head‑to‑head with token masking/asynchronous baselines. The paper critiques these strategies but does not supply a controlled replication (same data/compute) of a recent masking method (e.g., high‑entropy minority emphasis) to empirically validate “synchronous > masking/asynchrony.”

**Questions:**

1. How are token entropies computed and cached in practice—under the rollout policy or the updating policy—and how stable is token typing across epochs?

2. Do the same ε and β settings transfer to other backbones or to larger models, or is retuning required at new scales/architectures?

---

> ### Author Response · Authors · 2025-12-03
>
> We thank reviewer 2PyW for the positive support and insightful comments. We provide our point-wise responses below.
>
> ---
>
> ### **W1: "All results rely on a single 1.5B base (DeepSeek‑R1‑Distill‑Qwen‑1.5B); adding a second backbone or a larger‑scale model would strengthen generality. Further, The entropy quantile (ρ) is fixed without a sensitivity study, and the work would benefit from either a small sweep or an adaptive rule of thumb."**
>
> **A**:
>
> **(1) Backbone**: DeepSeek-R1-Distill-Qwen-1.5B is a strong and widely used backbone in recent RLVR work [1, 2]. This allows fair comparison with prior methods. We agree that larger backbones would improve generality. Due to limited compute, we cannot include them in this version, but we will add results on a 7B model in the final version.
>
> **(2) Entropy quantile**: We selected the 80th percentile based on prior work [1], which showed that focusing on high-entropy minority tokens is effective for reasoning tasks.
>
> ### **W2: "No direct head‑to‑head with token masking/asynchronous baselines. The paper critiques these strategies but does not supply a controlled replication (same data/compute) of a recent masking method (e.g., high‑entropy minority emphasis) to empirically validate “synchronous > masking/asynchrony.”"**
>
> **A**: Thank you for pointing this out. Our goal is to study synchronous updates, while masking and asynchronous methods break token dependencies. We will add a controlled comparison with a recent masking baseline under the same data and compute in the final version.
>
> ---
>
> ### **Q1: "How are token entropies computed and cached in practice—under the rollout policy or the updating policy—and how stable is token typing across epochs?"**
>
> **A**:
>
> **(1) Entropy computation**: For each step, we generate responses under the rollout policy with vLLM. Then we use the same model with FSDP for forward logits and entropy computation. The computed data is then used for policy optimization. This avoids inconsistencies between rollout and training.
>
> **(2) Stability**: Our experiments show that token typing remains stable across epochs. This matches the findings in [3], which finds that the overall entropy pattern (i.e., which tokens have high entropy and which have low entropy) remains largely consistent throughout RLVR training.
>
> ---
>
> ### **Q2: "Do the same ε and β settings transfer to other backbones or to larger models, or is retuning required at new scales/architectures?"**
>
> **A**: The hyperparameters $\epsilon$ and $\beta$ control the balance between exploration and knowledge preservation. We expect them to transfer across similar architectures, but we agree that some retuning may be needed for optimal performance. We will test their transferability on a larger model and include results in the final version.
>
> **References**
>
> [1] DeepScaleR: Surpassing O1-Preview with a 1.5B Model by Scaling RL. Notion Blog, 2025.
>
> [2] ProRL: Prolonged Reinforcement Learning Expands Reasoning Boundaries in Large Language Models. NeurIPS 2025.
>
> [3] Beyond the 80/20 Rule: High-Entropy Minority Tokens Drive Effective Reinforcement Learning for LLM Reasoning. NeurIPS 2025.

---

### Official Review · Reviewer_PKQW · 2025-11-10

**Soundness:** 2
**Presentation:** 2
**Contribution:** 1
**Rating:** 2
**Confidence:** 4

**Summary:**

The paper proposes Archer, an RL training scheme that treats tokens with high entropy as “reasoning” and tokens with low entropy as “knowledge.” It keeps updates synchronous but applies looser constraints (clip, KL) on high-entropy tokens and tighter constraints on low-entropy tokens. Experiments show gains on math (AIME24/25) and code (LiveCodeBench) with a 1.5B Qwen backbone.

**Strengths:**

1. The method is simple. Step-wise entropy is easy to compute and does highlight where models tend to struggle.
2. The paper gives ablations showing that removing KL on low-entropy tokens causes collapse.

**Weaknesses:**

1. The core assumption is that entropy reliably separates reasoning from knowledge, which is questionable. Entropy shifts with sampling, style, and prompt form, and generation length.
2. The paper uses some heuristics which needs more justification to claim effectiveness, such as the fixed empirical quantile threshold for entropy.
3. The evaluation is narrow (math, code) and does not test whether the method generalizes to less structured reasoning. I feel them not convicing since we all know the noise w.r.t randomness and, more importantly, fair comparison, in these settings.
4. Data hygiene is hand-waved; contamination remains a real issue in math tasks.
4. The KL story is incomplete: the paper shows collapse when $\beta=0$, but does not study adaptive KL or schedules that could match Archer’s behavior without entropy gating.

**Questions:**

see weakness

---

> ### Author Response · Authors · 2025-12-03
>
> We thank reviewer PKQW for the insightful comments. We provide our point-wise responses below.
>
> ---
>
> ### **W1: "The core assumption is that entropy reliably separates reasoning from knowledge, which is questionable. Entropy shifts with sampling, style, and prompt form, and generation length."**
>
> **A**: We agree that absolute entropy values change under different sampling settings. However, recent work [1] and our own experiments show that the **relative ranking** of token entropies inside a single response is **stable**. This stability is important because our method uses **response-level quantiles**, not absolute entropy values. Therefore the entropy threshold adapts to each response, which reduces the influence of sampling or style. This allows entropy to remain a reliable signal for separating reasoning and knowledge tokens.
>
> ---
>
> ### **W2: "The paper uses some heuristics which needs more justification to claim effectiveness, such as the fixed empirical quantile threshold for entropy."**
>
> **A**: Prior work [1] reports that the top 20 percent high-entropy tokens play a central role in reasoning. We follow this finding. Our experiments also confirm that selecting the 80th percentile at the response level gives stable results. We will clarify this motivation in the final version.
>
> ---
>
> ### **W3: "The evaluation is narrow (math, code) and does not test whether the method generalizes to less structured reasoning."**
>
> **A**: (1) Our evaluation setup follows previous work [1, 2, 3], which mainly focuses on mathematical and code reasoning. We agree that testing on less structured reasoning tasks is valuable and we will explore this direction in future work.
>
> (2) We also take strong measures to ensure fair evaluation. To reduce randomness, we report results using `avg@64` on AIME24, AIME25, and AMC23. Many recent works only report `avg@32`. This reduces the noise and gives a more fair comparison.
>
> ---
>
> ### **W4: "Data hygiene is hand-waved; contamination remains a real issue in math tasks."**
>
> **A**: (1) We perform strict contamination checks following DeepScaleR and Math-Verify pipelines. All training sets are cleaned by n-gram removal, duplication filtering across sources, and verification scripts. We will add a more detailed description in the final version.
>
> (2) There is some degree of data contamination in the Qwen models on older benchmarks such as MATH500, AMC23, and AIME24. This may be unavoidable when pre-training on large-scale web corpora that include publicly available benchmark problems, as noted in [1]. However, newer benchmarks like AIME25 and MinervaMath are likely free from such contamination. Importantly, our method consistently outperforms the baselines on both the older benchmarks (MATH500, AMC23, AIME24) and the newer ones (AIME25 and MinervaMath), demonstrating the robustness and effectiveness of our approach.
>
> ### **W5: "The KL story is incomplete: the paper shows collapse when $\beta=0$, but does not study adaptive KL or schedules that could match Archer’s behavior without entropy gating."**
>
> **A**: We agree that KL control can be adaptive. Our binary split on token type is a simple and effective choice supported by prior work [1]. Our experiments also show that using different constraints for the two types improves optimization. We will explore adaptive KL schemes or multi-level token splits in future work.
>
> **References**
>
> [1] Beyond the 80/20 Rule: High-Entropy Minority Tokens Drive Effective Reinforcement Learning for LLM Reasoning. NeurIPS 2025.
>
> [2] DeepScaleR: Surpassing O1-Preview with a 1.5B Model by Scaling RL. Notion Blog, 2025.
>
> [3] ProRL: Prolonged Reinforcement Learning Expands Reasoning Boundaries in Large Language Models. NeurIPS 2025.
>
> [4] Reasoning or Memorization? Unreliable Results of Reinforcement Learning Due to Data Contamination. arXiv 2025.

---

### Meta-Review · Area_Chair_qX7v · 2026-01-13

**Summary:**

The reviewers identified critical concerns centered on three main issues: (1) severely limited experimental scope with validation only on a single 1.5B model across math and code tasks, raising serious generalizability questions; (2) absence of direct comparisons with essential recent baselines like High Entropy Minority Tokens and Lopti, which is a fundamental experimental requirement; and (3) insufficient theoretical justification for key design choices including the binary entropy split and why synchronous dual constraints outperform existing approaches. Additional concerns include the introduction of multiple sensitive hyperparameters without adequate transferability analysis, potential data contamination issues, and over-reliance on heuristics from prior work without validation for this specific method.

**Reviewer Concerns:**

The rebuttal effectively addressed some technical clarifications, particularly regarding ablation study locations (Reviewer Hrnn) and entropy computation details (Reviewer 2PyW). However, critical concerns remain unresolved. The experimental scope limitation to a single 1.5B model across only math and code tasks was only met with promises of future 7B results rather than actual evidence. The absence of direct comparisons with essential recent baselines like High Entropy Minority Tokens and Lopti, raised by multiple reviewers, was acknowledged but deferred to the camera-ready version. The theoretical justification for why binary entropy splitting is sufficient and why synchronous dual constraints fundamentally outperform existing masking approaches was not substantively defended beyond citing prior work. Hyperparameter transferability concerns also received only partial responses without new validation, leaving the core methodological and experimental gaps unaddressed.

**Reviewer Scores:**

Reviewer PKQW would likely maintain their rejection or marginally increase to 3, as core methodological concerns remain unaddressed. Reviewers 2PyW and uydxwould probably maintain their scores, appreciating clarifications but noting absent key experiments. Reviewer Hrnn (4) would likely increase to 5-6 due to effective clarifications on ablations and PPO compatibility. Reviewer XjH3would probably decrease to 6-7, as their explicitly requested direct baseline comparisons were only promised rather than provided.

---

### Decision · Program_Chairs · 2026-01-26

Reject